# Dried Blood Spots as Matrix for Evaluation of Valproate Levels and the Immediate and Delayed Metabolomic Changes Induced by Single Valproate Dose Treatment

**DOI:** 10.3390/ijms23137083

**Published:** 2022-06-25

**Authors:** Sing Teang Kong, Hai-Shu Lin, Jianhong Ching, Huiqing Xie, Paul C. Ho

**Affiliations:** 1Department of Pharmacy, National University of Singapore, 18 Science Drive 4, Singapore 117543, Singapore; singteang@gmail.com (S.T.K.); linhaishu@sztu.edu.cn (H.-S.L.); 2College of Pharmacy, Shenzhen Technology University, Shenzhen 518118, China; 3Duke-NUS Medical School, 8 College Road, Singapore 169857, Singapore; jianhong.ching@duke-nus.edu.sg; 4KK Research Centre, KK Women’s and Children’s Hospital, Singapore 229899, Singapore; 5Institute of Materials Research and Engineering, Agency for Science Technology and Research, Singapore 138634, Singapore

**Keywords:** dried blood spot (DBS), gas chromatography/mass spectrometry (GC/MS), metabolomics, pharmacokinetics, valproate

## Abstract

The immediate and delayed metabolic changes in rats treated with valproate (VPA), a drug used for the treatment of epilepsy, were profiled. An established approach using dried blood spots (DBS) as sample matrices for gas chromatography/mass spectrometry-based metabolomics profiling was modified using double solvents in the extraction of analytes. With the modified method, some of the previously undetectable metabolites were recovered and subtle differences in the metabolic changes upon exposure to a single dose of VPA between males and female rats were identified. In male rats, changes in 2-hydroxybutyric acid, pipecolic acid, tetratriacontane and stearic acid were found between the control and treatment groups at various time points from 2.5 h up to 24 h. In contrast, such differences were not observed in female rats, which could be caused by the vast inter-individual variations in metabolite levels within the female group. Based on the measured DBS drug concentrations, clearance and apparent volume of distribution of VPA were estimated and the values were found to be comparable to those estimated previously from full blood drug concentrations. The current study indicated that DBS is a powerful tool to monitor drug levels and metabolic changes in response to drug treatment.

## 1. Introduction

Metabolomics is a scientific discipline that associates findings from the metabolite profiles to the disease state [1]. As abnormality from genetics and proteins may not always translate into clinically defining symptoms, fluctuations in metabolomics within biological cells and/or in biological fluids are often proposed to be the direct or causal indicators of some diseases [2,3]. It has been suggested that the metabolomic approach is well-suited for epilepsy studies, as it can capture the metabolomic changes after an epileptogenic insult and also possibly indicate the metabolomic responses to the treatment [4].

It is irrefutable that the best way to study metabolomics will be through the direct analysis of the tissues or biological fluids of the affected areas. For example, to study epileptogenesis, it is best to perform a direct analysis of the brain tissues or cerebrospinal fluid (CSF) of the affected subjects and then compare the findings to those of the non-affected subjects. In such, it will be the most informative study, outlining the pathways altered in the generation of epilepsy. However, due to the complexity of ethical issues involved in obtaining some of these valuable samples, research has previously been conducted using alternative matrices, such as animal brain tissues [5] and livers [6,7] as well as human plasma [8,9] and urine [10]. We previously demonstrated that dried blood spots (DBS) could be a potential substitution for conventional plasma as a sample matrix for metabolomic study using gas chromatography/mass spectrometry (GC/MS) [11]. DBS has the advantage of stabilizing metabolites in their dry forms, while being easily acquired through a minimally invasive technique (only a needle prick). However, when comparing the metabolomic profiles of the respective plasma and DBS matrix, DBS was unable to recover and identify several markers, such as L-lysine, iminodiacetic acid, DL-threo-beta-hydroxyaspartic acid, citric acid or adenosine-5-monophosphate. We hypothesize that the discrepancy could be due to a deficiency in the extraction of analytes from the DBS matrix. In this study, we attempted to improve the extraction efficiency and recover these metabolites from DBS through modifying the sample extraction process using double solvents. The modified approach was then employed for metabolomic profiling of rats treated with VPA.

Valproic acid (2-propyl-pentanoic acid, VPA) is a histone deacetylase inhibitor that belongs to the older generation of anticonvulsants [12,13]. It has been proven to be one of the most effective agents for a broad range of epilepsies, including primary generalized seizures and complex partial seizures [14]. Additionally, VPA has also received FDA approval for the treatment of manic bipolar I disorder and prophylaxis against migraine. The compound is a simple branched chain fatty acid that is highly protein bound (90% in human and 64% in rats) and predominantly metabolized via hepatic glucuronidation and mitochondrial β-oxidation.

So far, the mechanisms of actions of VPA are believed to be multifaceted, and not fully understood. VPA has been postulated to be able to enhance selectively the post synaptic brain gamma-aminobutyric acid (GABA) responses, acting directly on neuronal membranes, reduce aspartate excitatory transmission and inhibit reuptake of GABA into the glia and nerve endings [15,16]. In recent years, several studies with the respective metabolomics, proteomics and genomics approaches were able to reveal the VPA effects on metabolic changes such as inducing lipid abnormalities, perturbation in glycogenolysis and hyperinsulinemia [5,6,7,10]. However, these studies focused primarily on the mechanisms of VPA-induced hepatotoxicity [6,7,10]. Hence, the doses administered in these studies were high, averaging at 600 mg/kg [5,6,7]. The mechanisms involved in inducing toxicities may or may not represent the metabolic effects of VPA at therapeutic levels. As suggested by Dickinson et al., when plasma concentrations of VPA exceeded 100 µg/mL, there may already be saturations in some metabolism pathways [17]. Thus, the manifested metabolic changes could be of other pathways that have been activated by the unusual high doses.

A study by Kheder et al. has established a few clinical biomarkers, such as thrombocytes, amylase, C-reactive protein and potassium, that can be easily monitored to gauge patients’ responses to chronic VPA therapy [18]. In view of the complexity of VPA’s actions, we proposed to investigate the metabolite profiles resulting from treatment with VPA over a period of 24 h, using DBS as the sample matrix. To minimize the inherent metabolic changes due to the epilepsy disease itself, normal healthy rats were used. The dose of VPA used in this study was the lowest in the dosing range used by previous studies in the investigation of its toxicities [6,7,10]. The objective of this study is to capture snapshots of the immediate and delayed metabolite changes that may provide insights into the metabolic pathways involved in its mechanism of actions.

## 2. Results

### 2.1. Metabolomics

The metabolites were identified using the Agilent Fiehn GC/MS Metabolomics RTL (retention time locked) Library [19]. Clustering in principal component analysis (PCA) and partial least square discriminant analysis (PLS-DA) analysis are driven by the similarities in the metabolite profiles obtained. Visual inspection of the PCA plot found no discrimination between the control and treatment groups, with respect to the post-VPA administration for neither male nor female rats. After removing the time post-VPA treatment as a factor, partial clustering was observed for the control and treatment groups in the unsupervised algorithm, PCA loadings plot in Figure 1a. PCA can help to explore the linear relationships between data variables and between data samples. It usually works well if the variables are highly correlated. The partiality in clustering demonstrated in the figure indicated that some rats’ metabolite profiles resemble the other group more than its actual group (Figure 1a). In the subsequent supervised algorithm modeling, PLS-DA, clustering of the control and treatment group was demonstrated in Figure 1b. PLS-DA is a method for exploring cause and effect association via a regression model. The discrimination observed indicated that the differences in metabolite profiles of the control, as compared to the treatment groups, would ultimately determine their classification accurately.

Orthogonal PLS-DA (OPLS-DA) resulted in one predictive and six orthogonal (1 + 6) components. Its predictive performance Q^2^(Y) is 90.1%, while the total explained variance R^2^(X) is 38.4% (Figure 2). The original PLS-DA model that showed clear clustering between the control and treatment group was valid based on the validation plot, where the Q^2^ regression line gave a negative intercept, −0.246, while all the simulated R^2^ values were less than the original model value on the right (Appendix A).

Testing of the time-related fluctuations of the individual metabolite was then introduced for these compounds to identify differences between the treatment from the control for the respective male and female groups. The top five ranking compounds at each time point for male rats are shown in Appendix A.

In males, the ranking of compounds in order of their differences in relative changes in concentrations between the control and treatment groups revealed a putatively identified 2-hydroxybutyric acid at half an hour post-VPA treatment (Appendix A). Of note, 2-hydroxybutyric was also listed as the 10th most differentiating compound in the females at half hour post-VPA treatment. At 2.5 h post-VPA treatment, pipecolic acid was downregulated and ranked as the most important metabolite in differentiating between the control and treatment rats. Tetratriacontane was ranked second at the fifth hour and its level was upregulated in the post-VPA treatment rats. Stearic acid was upregulated from the 5th h to the 8th h in the VPA treated rats, ranking as the 4th and 5th most important metabolite.

In females, the top 5 ranking compounds at each time point from 0.5 h up to 24 h do not have any putatively identified metabolite, limiting further investigation. Therefore, they are not shown here.

Of interest, a look at the detectable compounds in DBS revealed the presence of L-lysine, iminodiacetic acid, DL-threo-beta-hydroxyaspartic acid, citric acid and adipamide, which were not extracted from DBS in our previous study [11], representing a limitation of DBS as a matrix for metabolomic study. The use of double solvents for extraction helped to recover these metabolites from DBS. However, the metabolite of adenosine-5-monophosphate 2 remained undetectable, even with this modified sample extraction method.

### 2.2. VPA Quantitation

Linearity of the VPA calibration curve was defined using 7 standard concentrations, ranging from 0.5 µg/mL to 5 mg/mL. The *R*^2^ of the calibration curve estimated from linear regression was found to be greater than 0.99. A total of 2 quality control (QC) samples at concentrations of 2 mg/L and 50 µg/mL were used. Their experimentally determined values were within 12% of the theoretical values and were consistent throughout the triplicates with coefficient of variation (CV%) of less than 5%.

The average drug concentrations in DBS measured from all VPA-treated rats are shown in Figure 3. The drug concentrations in females were consistently higher than those in males. However, generally, they displayed similar pharmacokinetic profiles; the drug concentrations declined rapidly, followed by a slight increase in concentrations that peaked at the 5th h post VPA administration.

Concentration-time profiles of VPA after intravenous injection revealed an average terminal elimination half-life (HL Lambda Z) of 1.86 ± 0.26 h and a mean residence time (MRT) of 1.30 ± 0.36 h, using a non-compartmental model from the pooled male and female treatment groups. However, when the male treatment group was compared to the female group, the clearance was found to be significantly higher in males (0.51 ± 0.12 L/hr) than in females (0.12 ± 0.01 L/h (*p* < 0.05), whereas the volume of distribution was also significantly larger in males (0.82 ± 0.23 L) than in females (0.12 ± 0.04 L) (*p* < 0.05). The pharmacokinetic parameters obtained are listed in Table 1.

Pearson correlation tests found insignificant correlations (*p* > 0.05, −0.7 < r < 0.7) between the individual endogenous metabolites with VPA concentrations (data not shown). Multiple linear regression analysis also did not reveal any important endogenous metabolites that can account for the measured VPA concentrations, as well as the inter-individual variations observed.

## 3. Discussion

In this study, we improved the recovery of some metabolites that would otherwise be undetectable by extracting the DBS with double solvents. By using solvents of different polarity, metabolites of diverse physical and chemical properties could be recovered. The modified method was adopted to discern the metabolic perturbations resulted from the VPA actions.

Time-related fluctuations in metabolite profiles were not observed in our study. This could be attributed to the fact that the times of blood collection mismatched the responses in metabolite levels, if any. Moreover, the natural circadian rhythm for the metabolite levels themselves [20] might have overshadowed the changes in response to the drug treatment. In addition, the fluctuations of each metabolite at the chosen time points in our study could be too subtle to be distinguished. The inter-individual variations may have annulled the effect of any intra-individual variations or vice versa. As the consequence of the seemingly unrelated rise and fall of each metabolite within one time point, PCA analysis was unable to correlate the profiles to their respective timings. These may have contributed to the non-observable differentiation in metabolite profiles according to their time of collection post-placebo or post-VPA treatments.

When the metabolite profiles of the rats were grouped regardless of the timing of collection, the profiles of the control rats were found to be different from the treatment rats, indicating that the discrimination was specific to VPA treatment. The difference was also found to be gender-specific with no overlap of the top 5 metabolites noticed. To discern the perturbations of metabolite levels attributed to VPA treatment; whether they are immediate or delayed, timing of the sample collections was reintroduced in the analysis using a three-feature ranking algorithm. The onset of metabolite changes was assumed to be reflected by the timing of the blood collection post-VPA treatment.

Most metabolomic studies conducted in recent years focused on biomarkers of hepatotoxicity or adverse events induced by VPA treatment [6,7,10]. The doses used in these studies were much higher (approximately 600 mg/kg) than ours (200 mg/kg). Nevertheless, in the study by Price et al. using residual urine from pediatric epilepsy patients as samples, the patterns of metabolism and excretion of organic acids were found to be altered after stable administration of VPA for at least 2 weeks [10]. Out of the many affected organic acids recovered from the urine samples in their study, only 2-hydroxybutyric acid was detected to be significantly increased at the 0.5 h post-VPA treatment in our study using DBS as matrix, but the elevated level was quickly resolved thereafter.

The present study is an untargeted metabolomic study; metabolites are screened without prior knowledge to the chemical identities of metabolites present. As such, internal standards were not used comprehensively at the start of this experiment; and it was also impossible to quantify all identified metabolites with confidence. Nevertheless, the repeatability of the assay has been established from the pooled quality control (QC) samples (Appendix A); and the accuracy of detection has been confirmed through quantifying samples spiked with analytes with different lipophilicity at low and high concentrations (Appendix A). The procedures were performed according to the guidelines for mass spectrometry assays applied in untargeted metabolomic studies [21]. We acknowledge that another limitation is that the use of caffeine as internal standard in the present study may lead to inaccurate measurements of some very lipophilic metabolites. Due to the deficiency, the etiologies of the observed metabolic changes are not further elaborated. In the further targeted metabolomic study, the metabolites defined in the study can be quantified using suitable internal standards in the assay.

VPA concentrations measured from the whole blood (DBS) demonstrated a typical pharmacokinetic profile of enterohepatic recirculation, which has been demonstrated in previous studies [17,22]. There was an initial steep decline in drug concentration due to rapid elimination, followed by a secondary increase at approximately the 5th h and a slower elimination thereafter. Male rats in our study were observed to have lower concentrations throughout, demonstrating faster clearance compared to their female counterparts. As in our study, the male rats in our study had an overall greater body weight than the female rats of the same age (317.4 ± 22.03 g for male, 213.8 ± 15.91 g for female); the differences in the pharmacokinetic profiles of VPA between male and female rats could be due to the smaller liver weight in females, which was still smaller in females even after normalized to their body weights, as demonstrated in a previous study [22]. However, it remains inconclusive if liver size could play a substantial role in its biotransformation [23]. 

The VPA terminal half-life (t_1/2_) determined in this study was comparable to the plasma half-life reported in a study by Binkerd et al. [24]. In their study, female rats were given 200 mg/kg of VPA via oral gavage. Their t_1/2_ value was 1.0 ± 0.2 h. However, both of their AUC and maximum concentration (C_max_) at 0.5 h, which were 1019 ± 769 µg·h/mL and 341 ± 18 µg/mL, respectively, were much higher than that estimated in our study [24]. It is important to note that the VPA concentrations reported in their study were measured from plasma, while ours were estimated from DBS levels. Plasma/whole blood concentrations ratio of VPA had been demonstrated to be 1.46 ± 0.03 (mean ± standard error) [17]. Therefore, it is not surprising to observe the differences.

Comparisons were also made between our study and the study by Dickinson et al., where the pharmacokinetic parameters were estimated using whole blood concentrations [17]. The male rats in their study were dosed at 150 mg/kg. The t_1/2_ was found to be 0.68 h, which was much lower than ours. However, when the rats were dosed at a much lower concentration of 15 mg/kg, the reported values for CL (0.525 ± 0.065 L/h) and V_d_ (0.155 ± 0.025 L) were close to that determined in our study.

## 4. Materials and Methods

### 4.1. Chemicals and Reagents

Methanol, chloroform, acetonitrile and hexane of analytical grade were purchased from Prime Products Pte Ltd. (Singapore). Caffeine was purchased from Sigma-Aldrich (St. Louis, MO, USA), N-methyl-N-trimethylsilyltrifluoroacetamide (MSTFA) with 1% trimethylchlorosilane (TMCS) from Thermo Scientific Pte Ltd. (Waltham, MA, USA) and sodium hydroxide from JT Baker (Phillipsburg, NJ, USA). The internal standard used in VPA quantitation, 3-methoxy phenylmethylene hydantoin (3MP), is a chemical derivative of phenytoin, courtesy of Dr Wai Keung Chui, Medical Chemistry Program Research Group (National University of Singapore). Deionized water was obtained from a Milli-Q system (Millipore, Boston, MA, USA) and used throughout the experiments.

### 4.2. Animals

This in vivo study was performed according to the ‘Guidelines on the Care and Use of Animals for Scientific Purposes’ (National Advisory Committee for Laboratory Animal Research, Singapore). The animal handling procedures were reviewed and approved by the Institutional Animal Care and Use Committee of the National University of Singapore (NUS) (approval number: 001/12, 14 February 2012). This animal model had been applied in previous metabolomic studies [3,11].

Sprague-Dawley rats (7–8 weeks old, males: 10, females: 10) were obtained from the Comparative Medicine Center (CMC) of NUS. The rats were housed in a specific pathogen free animal facility (24 °C, 60% relative humidity) at CMC and kept on a 12-h light/dark cycle with free access to food and water. Twenty-four hours before the study, a polyethylene tubing (I.D. 0.58 mm, O.D. 0.965 mm, Becton Dickinson, Sparks, MD 21152, USA) was implanted into the right jugular vein under isoflurane anesthesia. This catheter was used for drug injection and blood sampling [25,26].

At 9 am on the study day, five rats from each gender were administered VPA intravenously at a dose of 200 mg/kg body weight (treatment group). The rest of the 5 rats from each gender were given sodium chloride 0.9% intravenously, as the placebo control group. Prior to VPA administration, blood samples (approximately 300 µL) from each of the 10 male rats and 10 female rats were collected and kept in an ice box until they were processed. After VPA/saline injection, blood samples were collected at 0.5, 2.5, 5, 8 and 24 h, respectively, from the 20 rats. The 6 time points were anticipated to capture the metabolite fluctuations in a day. Within 30 min of collection, the blood samples were spotted in duplicates of 25 µL onto Guthrie paper (UK Neonatal Screening Laboratories Network, Whatman 903^®^) and were left to dry in the fume hood for at least 3 h, the minimum time required for complete dryness [11]. After drying, the DBS were punched, and the cores (6 mm) obtained were placed inside the respective Eppendorf tubes. All the acquired DBS were stored at −80 °C until processing for analysis [27,28]. One DBS was used for metabolomic analysis and the other one was used for drug level quantification.

### 4.3. Sample Processing for Metabolomic Analysis

DBS was extracted twice, first using 150 µL of methanol and subsequently using 150 µL of chloroform, with caffeine as the internal standard at a concentration of 6.25 ng/mL in the extraction solvent. After addition of each solvent, the sample was vortexed for 20 min. The pooled methanol and chloroform extracts were centrifuged at 21,000× *g*, 22 °C for 10 min and 240 µL of the resulting supernatant was transferred to 15 mL Kimble centrifuge glass tubes (Gerresheimer Glass AG, Düsseldorf, Germany) for evaporation. The sample was dried at 40 °C under a stream of nitrogen. Thereafter, 100 µL of toluene was added. After vortexing for 15 s, the sample was dried again under similar conditions. Derivatization of the dried sample was completed using 100 µL MSTFA with 1% TMCS and incubation at 60 °C for 1 h. The products of derivatization were centrifuged at 3500× *g*, 22 °C for 10 min before 90 µL of the final supernatant was transferred into a 200 µL conical base inert glass insert, placed in a 2 mL amber autosampler glass vial (Agilent Technologies, Waldbronn, Germany).

### 4.4. Sample Processing for VPA Quantitation

Extraction of VPA from DBS was carried out using 480 µL of acetonitrile and 20 µL of 1 N NaOH, vortexed for 1 min and followed by sonication for 5 min. The resulting extracts were centrifuged at 3500× *g*, 22 °C for 15 min [29,30]. An amount of 400 µL of the supernatant were transferred into 15 mL Kimble centrifuge glass tubes for drying under a stream of nitrogen at 30 °C, followed by the removal of water using toluene and drying under nitrogen again. The dried sample was then derivatized using 50 µL MSTFA with 1% TMCS with incubation at 70 °C for an optimum period of 50 min. Dilution was achieved with 50 µL of hexane. After vortexing for 1 min, 80 µL of the final volume was transferred into a 200 µL conical base inert glass insert, placed inside a 2 mL amber autosampler glass vial.

### 4.5. GC-MS Settings for Metabolomic Analysis

Analyses were carried out using the 7890A GC System that was coupled to 5975 inert MSD with Triple-Axis Detector (Agilent Technologies), using its in-built Fiehn method of analysis for metabolomic screening [19].

### 4.6. GC-MS Settings for VPA Quantitation

An assay for quantification of VPA concentrations in DBS was optimized in accordance to the International Conference of Harmonization (ICH) guideline. The settings were as follows: an injection temperature of 200 °C, ion source temperature of 220 °C, split ratio of 1:5, the starting temperature set at 90 °C for 0.2 min, with initial gradient ramp at 10 °C/min to 120 °C and held for 0.5 min. The temperature was then increased at 65 °C/min up to 285 °C and held for another 0.5 min. The temperature ramp was then reduced to 10 °C/min to reach a temperature of 291 °C and held for 0.2 min, before continuing at 60 °C/min to reach a final temperature of 300 °C, with a hold at 300 °C for 5 min. Selective ion monitoring (SIM) mode was applied for identifier ions with an m/z value of 201 and qualifying ions with m/z values of 145 and 129. A calibration curve was established with 6 concentrations, ranging from 0.5 µg/mL to 5 mg/mL.

### 4.7. Statistical Analysis

Automated Mass Spectral Deconvolution and Identification System (AMDIS) 32 version 2.66, 2008 was used as the deconvolution software. A minimum match factor of 45%, adjacent peak subtraction of one, medium resolution of the peaks with high sensitivity and medium shape requirements were used as the deconvolution settings. Simple mode matching was carried out against their putative identities with Fiehn library (with the addition of caffeine).

The absolute peak areas for the compounds were normalized against that of the internal standard. Compounds that were present in less than two samples from the same group (either in the control or treatment group) were removed. The total number of compounds without much information was reduced from 3370 to 1311. Using each rat as its own control, the normalized peak areas for each compound at time 0 was subtracted from the normalized peak areas for the compound in the same rat at other time points (i.e., 0.5, 2.5, 5, 8 and 24 h).

The data list was then subjected to chemometric analyses using Soft Independent Modelling of Class Analogy (SIMCA) P+ version 12.0.1 (Umetrics, Sweden). The analyses included principal component analysis (PCA) and partial least square discriminant analysis (PLS-DA) and orthogonal PLS (OPLS). DBS were defined as their respective hours’ post-VPA administration. The DBSs were also classified into control and treatment group. The score plots generated were visually examined for clustering trends and outliers. Permutations with 100 iterations were utilized to investigate the model predictability and over fitting.

The respective feature ranking algorithm such as chi-squared, Relief-F and Information Gain, was used to rank the compounds in order of their differences in relative changes in concentrations between the control and treatment group for each sex at each time point. A consensus list was then derived by summing the normalized weights determined by these algorithms for each compound.

Additionally, the discriminating compounds between the control and treatment groups were correlated and regressed using Statistical Package for the Social Sciences (SPSS) version 11.0. Microsoft Office Excel was used for other compilation.

Validation of the assay for the metabolomic studies on the metabolite stability during storage, detection precision and recoveries is shown in Appendix A.

### 4.8. Pharmacokinetic Analysis

Concentration-time profiles of VPA after intravenous bolus injection were fitted into a non-compartmental model (WinNonlin Professional, Version 5.1, Pharsight Corporation, Mountain View, CA, USA). The pharmacokinetic parameters obtained included the area under the DBS concentration versus time curve (AUC_0–8 h_), apparent volume of distribution (V_d_) clearance (Cl), terminal elimination half-life (HL Lambda Z) and mean residence time (MRT) of VPA.

## 5. Conclusions

In summary, this study has shown that it is possible to use DBS for untargeted metabolomic study and drug quantification. The metabolic changes in response to drug treatment could then be monitored. To accomplish these objectives, metabolites and/or drugs of varying physical and chemical properties can be extracted from DBS through using combined extraction solvents of different polarity. In this study, we observed a reduction in pipecolic acid levels in male rats at 0.5 h after VPA treatment and increment of 2-hydroxybutyric acid at 2.5 h, tetratriacontane at 5 h only and stearic acid from 5 h to 8 h. Currently, there does not appear to be any immediate metabolite changes that reflect the effects of VPA treatment in the epilepsy control, aside from the metabolites that reflect toxicity. It is yet to be confirmed whether the changes in these levels after VPA treatment are due to causal or spurious association. These postulations could be confirmed in further studies by assessing the metabolite profiles in the respective subjects on chronic doses of VPA responding and resistant to the treatment. In this regard, DBS is a very useful approach as it is less invasive process to monitor drug levels and simultaneously profile the metabolic changes in response to drug treatment in a large patient population.

## Figures and Tables

**Figure 1 ijms-23-07083-f001:**
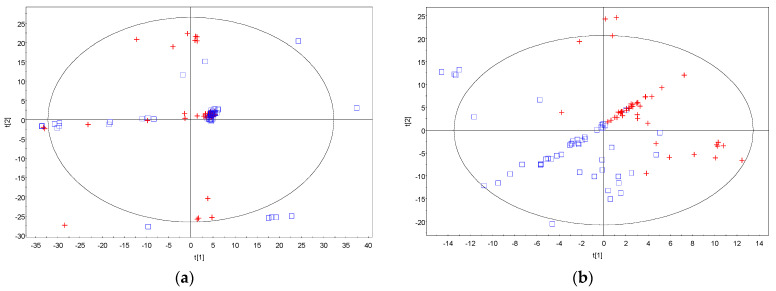
Principal component analysis and partial least square discriminant analysis. (**a**) Principal component analysis (PCA) loadings plot showing partial clustering of the control group (blue square) and the VPA treatment group (red cross); (**b**) partial least square discriminant analysis (PLS−DA) loadings plot showing clustering of the control group (blue square) and the VPA treatment group (red cross).

**Figure 2 ijms-23-07083-f002:**
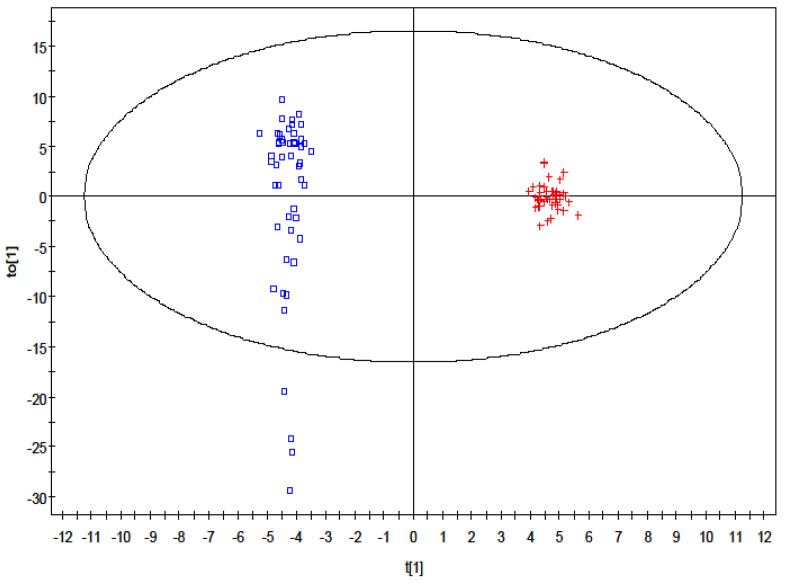
Orthogonal partial least square discriminant analysis loadings plot showing distinctive clustering of the control group (square) and the VPA treatment group (cross).

**Figure 3 ijms-23-07083-f003:**
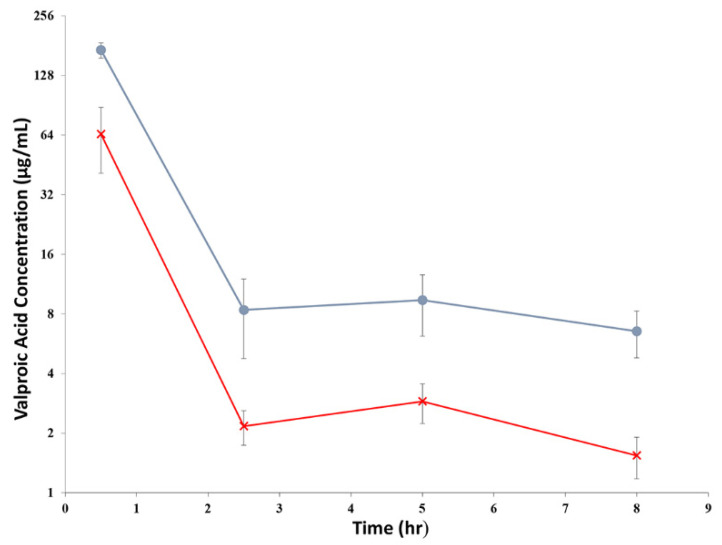
The of average VPA concentrations, μg/mL (mean ± S.D., *n* = 10) in DBS in rats after given an intravenous dose of 200 mg/kg. The circle (●) represented the concentrations in female rats, while the cross (×) represented the concentrations in male rats.

**Table 1 ijms-23-07083-t001:** Pharmacokinetic parameters estimated from the DBS drug levels following intravenous bolus administration of VPA in rats at dose of 200 mg/kg. Mean ± SD (*n* = 10).

Pharmacokinetic Parameters	Male	Female	All
Area under the curve (AUC_0–8 h_), µg·h/mL *	126.08 ± 39.20	363.26 ± 18.69	244.67 ± 132.78
Clearance (CL), L/h *	0.51 ± 0.12	0.12 ± 0.01	0.32 ± 0.23
Volume of distribution (V_d_), L *	0.82 ± 0.23	0.21 ± 0.01	0.51 ± 0.36
HL Lambda Z, h	1.71 ± 0.18	2.01 ± 0.24	1.86 ± 0.26
Mean residence time (MRT), h	1.14 ± 0.28	1.46 ± 0.40	1.30 ± 0.36

* denotes significant difference (*p* < 0.05) was found between the male and female treatment group.

## Data Availability

The data presented in this study are available on request from the corresponding authors.

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
