# Peer review of "Dried Blood Spots as Matrix for Evaluation of Valproate Levels and the Immediate and Delayed Metabolomic Changes Induced by Single Valproate Dose Treatment"

_ijms, 2022, doi:10.3390/ijms23137083_

Round 1

Reviewer 1 Report

Authors Kong et al. present a thoughtful set of experiments to assess levels of some key metabolites in response to valproate (VPA) treatment. The authors describe recovery of VPA-associated metabolites from DBS samples and describe a double extraction processing and GC-MS analysis protocol. Additionally, the authors perform some chemometric analyses to view gross metabolic changes between treatment and control groups. This is a good study, which deserved to be published in IJMS. There are, nevertheless, a few major and minor revisions that I suggest the authors make to improve the quality of their manuscript.

Major Revisions:

1.     Since the Q2 intercept is negative, see lines 128, this model is overfit. This is not a fatal error, but the authors do not discuss how this may impact their findings or future use of such a model. Please add.

2.     Since the authors are also providing a protocol for metabolite recovery from DBS, they should report the analytical performance of the method. Although some measure of QC testing is provided, the authors should also report inter- and intra-day CVs using a QC sample. They should analyze this sample 3x/day for 3 days and report %CV. These are minor experiments that should not take long and are worth the analytical burden. 

3.     Figure 3: Please perform significance testing within male and female rats. Although males and females appear to be significantly different at all timepoints, there does not appear to be a significant change in either males and females between ~2.5 hr and 8 hr. There appears to be a significant change for both males and females between 0.5 hr and 2.5 hr, but probably not after 2.5 hr. This is important as it means VPA is rapidly metabolized and that time estimations following VPA treatment can be reliably inferred only up to ~2.5 hrs.  

4.     Table 1: AUC is commonly reported from between 0-1.0 or as a corresponding percentage. Why have the authors listed AUCs in the hundreds? Please make sure this is correct.

Minor Revisions:

1.     On the title page, list the keywords in alphabetical order.

2.     For certain metabolites, names are identified with a ‘2’ following. For instance, on lines 56 and 57, we see ‘L-lysine 2, iminodiacetic acid 2,’ and ‘adenosine-5-monophosphate 2’. These numbers following metabolite names appear elsewhere as well, see lines 138, 147, and 151. What is the purpose of numbering them? It is confusing to me. Please either describe why these are numbered with ‘2’ (if necessary to include) otherwise, just delete them.

3.     Line 77: ‘glycogenoysis’ is misspelled and should be ‘glycogenolysis’.

4.     Line 103: “Hence, no PLS-DA plot was built”. Please delete this as you did obviously construct a PLS-DA plot (see Figure 1b).

5.     Figures should be listed after their first mention. Please re-order them to appear immediately after the paragraph in which they are first mentioned.

6.     Performance metric of the OPLS-DA, such as Q2(Y) and R2(X) should be italicized (lines 125 and 126, respectively). Also, please change Q2 and R2 to Q2 and R2 on lines 127 and 128, respectively.

7.     Line 137: change ‘compounds’ to ‘compound’.

8.     Line 155: change ‘r2 to ‘R2’.

9.     I think the structure of the paper would be more cogent if the authors placed the materials and methods before the results section, as the latter is informed by the former. Please switch.

In conclusion, this is a valuable study that not only builds logically from previous work but may facilitate related research on VPA and epilepsy treatment. The hypothesis is compelling, the methods are sound, and the results are clearly presented. Following appropriate responses to the recommended revisions above, this paper will be suitable for publishing in IJMS.

Author Response

Please refer to the file attached. Thanks

Reviewer 2 Report

Dried Blood Spots as Matrix for Evaluation of Valproate Levels and the Immediate and Delayed Metabolomic Changes Induced by Single Valproate Dose Treatment

S T Kong, HS Liu, J Ching, H Xie, and P C Ho,  Dept. Pharmacy, National Univ. Of Singapore

Overall comments:

The organization of the; paper could be improved. Following the introduction, suggest that the sections describing the materials and methods and statistical and pharmaceutical analyses should come next before the results and discussion sections.

Consider replacing the following words: Line 29, “formidable” with powerful

Line 41, “also probably” with possibly

Line 48: “continuously” with previously

Line 66, “also rendered” with received

Line 293, “a short chain fatty acid” with a long chain fatty acid; and following lines where “SCFA” is used. Replace with LCFA. (Stearic acid is C18:0 fatty acid. Short chain fatty acids are C2:0 to C8:0. Medium chain fatty acids are C10:0 to C14:0. And long chain fatty acids are C16:0 to C20:0 in chain length.)

Line 341, “lesser: with lower

Method: Measurements of immediate and delayed metabolic changes in rats treated with valproate (VPA) in dried blood spots; the metabolomics profiling was done following double solvent extraction, first with 150ul of methanol, and subsequently using 150ul chloroform with caffeine as the internal standard. After each solvent was added the DBS and solvent was vortexed for 20 min. The pooled methanol and chloroform extracts were centrifuged at 21,000 g, 22 °C., for 10 minutes, and 240ul of the supernatant was transferred to a 15ml glass tube and evaporated under nitrogen at 40°C. 100ul of toluene was added to remove trace amounts of water and the sample was dried again under Nitrogen. Samples were derivatized with MSTFA and TMCS and compounds were analyzed by GCMS.

Question 1: What was the amount of DBS sampled for the metabolomic study?

2: Why was caffeine chosen as the internal standard for the metabolic changes? Suggest that the authors could have added internal standards that are commonly used for quantitation of 2-hydroxybuttryic acid, pipecolic acid (d9-pipecolic acid), tetratriacontane, stearic acid (d3-C18:0), L-lysine-2, iminodiacetic acid, DL-threo-beta-hydroxy aspartic acid, citric acid and adipamide. In the discussion, 2nd paragraph, the statement that “the fluctuations of each metabolite at a specific time point...” Perhaps if the best internal standard for each metabolite had been used, there would have been better quantitation of each metabolite, and thus PCA analysis could have been done.

3. What amount of DBS was extracted for VPA quantitation? Please confirm the use of 3-methoxy phenyl methylene hydantoin as the internal standard and define the ions used for select ion monitoring (SIM) for the quantitation of VPA.

Author Response

(The authors gave the same response as above.)

Reviewer 3 Report

The authors attempted to conduct a non-targeted metabolomic analysis of the dried blood spots of the rats after valproate administration. The overall impression is not satisfactory. The following points should be discussed and improved by authors:

1) The table S1 is misleading as it contains strange signs, which apparently means the retention time and the main m/z value of the unknown metabolites. This table should be chahged.

2) Only 4 from 25 metabolites were identified. Their pure standards were not analysed using the described sample preparation methods and the reproducibility of their detection was not revealed. Thus, two conclusions could be maid: the modified method did not provide satisfactory results as it did not allow the authors to identify most of the significant metabolties; the changes in the level of identified metabolites could be artifacts. Considering the conclusions drawn, the discussion of the results in Section 3 seems premature.

3) Stearic acid is not SCFA as it contains 18 carbon atoms (line 293).

4) Some parts of the text are repeated (lines 170-176 &180-188; 425-429 & 453-459

Author Response

(The authors gave the same response as above.)

Round 2

Reviewer 2 Report

Dried Blood Spots as Matrix for Evaluation of Valproate Levels and the Immediate and Delayed Metabolomic Changes Induced by Single Valproate Dose Treatment

S T Kong, HS Liu, J Ching, H Xie, and P C Ho,  Dept. Pharmacy, National Univ. Of Singapore

Overall comments for the revised version of the paper:

The paper was not greatly improved from the previous version. The following suggestions were not done:

1) The organization of the; paper could be improved. Following the introduction, suggest that the sections describing the materials and methods and statistical and pharmaceutical analyses should come next before the results and discussion sections.

2)  The fact that the quantitation of the following metabolites was not improved by using the appropriate internal standards is a serious experimental design flaw.  

 Suggest that the authors could have added internal standards that are commonly used for quantitation of 2-hydroxybuttryic acid, pipecolic acid (d9-pipecolic acid), tetratriacontane, stearic acid (d3-C18:0), L-lysine-2, iminodiacetic acid, DL-threo-beta-hydroxy aspartic acid, citric acid and adipamide. In the discussion, 2nd paragraph, the statement that “the fluctuations of each metabolite at a specific time point...” Perhaps if the best internal standard for each metabolite had been used, there would have been better quantitation of each metabolite, and thus PCA analysis could have been done.

Author Response

Please refer to the file attached.

Reviewer 3 Report

Thanks to the authors for their response. However, I have to stick with my original decision. The absence of additional samples of discs with DBS for the target analysis indicates an incorrect experimental design. Judging by the purpose of the work and the volume of the Discussion section, it was metabolic profiling that was set as the main task of the work, and not the measurement of the level of valproate. In this case, the inability to check potentially significant markers is a serious flaw in the work.

The answer of the authors regarding reproducibility for other compounds than the target, unfortunately, cannot be used as an excuse. Based on my personal experience in the analysis of biological samples by GC-MS, I can say with confidence that it is unlawful to extend the determination patterns identified for specific compounds to other compounds, including related ones. For example, indolic acids differ in that their primary homomol can be determined under certain conditions, but its subsequent homologues under the same conditions are determined incorrectly. Similarly, succinic and fumaric acids can be reliably determined under certain conditions, but not α-ketoglutaric acid. There are many other examples.

Author Response

Please refer to the file attached.

Round 3

Reviewer 3 Report

The authors made the necessary corrections to the text of the manuscript, which made it possible to avoid unfounded conclusions at this stage of the study. In this form, the article can be published. At the same time, the significance of the results for publication in this journal should be assessed by the editors.